# A Rapid and High Throughput MIC Determination Method to Screen Uranium Resistant Microorganisms

**DOI:** 10.3390/mps3010021

**Published:** 2020-03-03

**Authors:** Meenakshi Agarwal, Rajesh Singh Rathore, Ashvini Chauhan

**Affiliations:** Environmental Biotechnology Laboratory, School of the Environment, 1515 S. Martin Luther King Jr. Blvd., FSH Science Research Center, Florida A&M University, Tallahassee, FL 32307, USA; rajeshsingh1.rathore@famu.edu (R.S.R.); ashvini.chauhan@famu.edu (A.C.)

**Keywords:** minimum inhibitory concentration (MIC), uranium (U), resistance, bacteria, yeast, fungi

## Abstract

The assessment of minimum inhibitory concentration (MIC) is a conventional technique used for the screening of microbial resistance against antibiotics, biocides, and contaminants such as heavy metals. However, as part of our ongoing work, we have observed biases associated with using traditional liquid MIC method to screen microbial heavy metal resistance, including both bacterial and fungal strains. Specifically, the addition of uranium into synthetic media causes immediate precipitation prior to the initiation of microbial growth, thus hampering the optical density measurements, and the obtained MIC values are thus flawed and inaccurate. To address this discrepancy, we report the optimization and development of a serial-dilution-based MIC method conducted on solid growth media supplemented with uranium, which is more accurate, relative to the testing of MICs performed in liquid cultures. Notably, we report on the efficacy of this method to screen not only bacteria that are resistant to uranium but also demonstrate the successful application to yeast and fungal isolates, for their ability to resist uranium, is more accurate and sensitive relative to the liquid method. We believe that this newly developed method to screen heavy metal resistance, such as uranium, is far superior to the existing liquid MIC method and propose replacing the liquid assay with the solid plate MIC reported herein.

## 1. Introduction

The release of toxic heavy metals into the environment continues to present serious threats to health and ecosystem-level processes [1]. Heavy metals, such as uranium (U), continue to accumulate in the environment and eventually contaminate the food web, presenting serious public health concerns [1,2,3]. Despite the toxicity imposed by exposure to heavy metals, microorganisms like bacteria, yeast, and fungi have recruited molecular mechanisms to not only tolerate or resist heavy metals, but also reduce environmental toxicity by immobilization of the metals [4,5,6,7,8,9,10]. Thus, microbially mediated decontamination of the environment is a promising approach [11,12]. Several groups are focusing on the isolation and characterization of metal-resistant microorganisms present in the environment [9,13,14,15,16] and largely depend on screening the microbial resistance, using the minimum inhibitory concentration (MIC) method, which is performed in various ways [9,17,18] for e.g., broth dilution, agar dilution, etc. In the broth dilution method, microorganisms are inoculated into a liquid growth medium in the presence of different concentrations of an antimicrobial agent. Microbial growth is assessed by using spectrophotometric cell counts after incubation for a specified time. The MIC value exists as the lowest concentration of the chemical which suppresses the microorganism’s growth [4]. In the case of the agar dilution method, a suspension with a defined number of purified microorganisms is directly spotted on a nutrient agar plate that has different concentrations of the test agent. Colony-forming units (CFUs) on the agar plate indicate the growth of that microorganism. Both methods include a series of dilutions, ranging from micro to macro, depending on the volume size. Organisms possessing appropriate genomic and cellular mechanisms to resist toxic compounds are typically found to respond with a higher MIC value when tested in the presence of the contaminant. Thus, the determination of MIC has become a gold-standard method for the surveillance of microbial resistance against several toxic compounds.

However, as part of an ongoing project that focused on the isolation and characterization of U-resistant bacteria and fungi from historically contaminated soils, we observed that the addition of U into the growth media caused precipitation of U, resulting in media turbidity not caused by growth of the isolate under study. It has been demonstrated earlier that many metals show the tendency of a pH-dependent speciation. The introduction of a metal into a buffered solution can lead to metal precipitation due to a change in the pH, presumably due to the acidic, neutral or slightly basic conditions. Precipitation may also depend on the metal complexation by an ingredient of the growth media used to grow the microorganism [19,20,21]. Regardless, the growth media becomes turbid causes interference with the measurement of bacterial growth in the presence of metal (e.g., U), resulting in inaccurate estimations of the ability of microorganisms to grow; thus, the obtained MICs are potentially flawed. To overcome these limitations, we have developed a quick and high-throughput method for determining MIC against U and potentially other heavy metals. In this study, we used a combination of microdilution and agar dilution methods, which we call the plate MIC method. This method permitted the screening of numerous microorganisms, including bacteria, yeast, and fungi, simultaneously, and the outcome could be visualized within an overnight incubation period. Therefore, this new method is rapid, sensitive, and more accurate compared to existing methods of heavy-metal-resistance determination and can serve as a powerful tool to evaluate microbial resistance against heavy metals that precipitate when used to supplement liquid growth media.

## 2. Materials and Methods

### 2.1. Identification of Microorganisms

Strains were isolated from soil samples obtained from the Savannah River Site (SRS) location, as described previously [9,16]. Patches of the pure isolates were made on Luria-Bertani (LB) agar, Yeast Extract-Peptone-Dextrose (YPD) agar and Potato Dextrose Agar (PDA) media for bacterial, yeast, and fungal isolates, respectively; all the strains were subjected to DNA extraction, using the ZR fungal/bacterial DNA kit (Zymo Research, Irvine, CA, USA). The 16S rRNA gene sequencing identified the strains as SRS-2-W-2017 (*Serratia* sp.), SRS-11-W-2017 (*Pseudomonas* sp.), MA-5-S-2018 (*Stenotrophomonas* sp.), SRS-9-S-2018 (*Serratia* sp.), and SRS-19-S-2018 (*Lysinibacillus* sp.). Using 18S rRNA gene sequencing, we identified the strains as SRS-4-S-2018 (*Aureobasidium* sp.), SRS-40-S-2018 (*Penicillium* sp.), and SRS-17-S-2019 (*Fusarium* sp.).

### 2.2. Plate Preparation

For bacterial strains, the LB agar medium was prepared, autoclaved, and allowed to cool down to 50 °C. Then, a 1 M stock solution of uranyl nitrate (Electron Microscopy Sciences) was prepared in sterile water. A required concentration of U (from 0 to 7 mM) was added from the stock, and the plates were poured. YPD (Yeast Extract Peptone Dextrose) and PDA (Potato Dextrose Agar) medium were used for yeast and fungi, respectively.

### 2.3. Plate MIC Method

All the tested microbial strains were patched on their respective nutrient agar plate. The following steps were carried out and described in a day-wise manner (Figure 1A).

#### 2.3.1. Day 1

A small loopful of inoculum was taken from the growing patches of microbes on the agar plate and added to a microfuge tube containing 1 mL of sterile water. Each strain was added in a separate microfuge tube, in a similar way, and properly mixed by vortex to form a homogeneous suspension.

Using a 96-well microtiter plate, we added 270 µL of sterile water to each well, using a multichannel pipette.

The prepared bacterial suspension of 0.1 OD was added to the first well of lane 1 on the microtiter plate, and each strain was added consecutively, from top to bottom. To make the dilutions series, 30 µL of suspension was transferred from the first well to the next and was repeated until we reached a dilution of 10^−7^. This produced the dilution ratio of 1:10.

With the help of a multichannel pipette, 2.5 µL of suspension was placed on LB plates that were prepared with different concentrations of U, ranging from 0 to 7 mM. A 20 mm diameter plate can accommodate 6–8 strains and their dilution series, as shown in Figure 1.

The plates were sealed with parafilm and incubated at 30 °C.

#### 2.3.2. Day 2

Growth was observed on the plate, and the MIC value was recorded for each strain. Photographs were taken for the record (Figure 1B).

### 2.4. Broth Dilution Method

The following steps were carried out.

#### 2.4.1. Day 1

An individual strain was inoculated into a culture tube, having 5 mL of LB broth, and was grown overnight, at 200 rpm, at 30 °C.

#### 2.4.2. Day 2

Culture tubes containing 5 mL of LB broth were prepared, and different concentrations of U were added in each tube. A test strain grown overnight was inoculated at 0.1 OD in all the prepared tubes and allowed to grow for the next 24 h, at 200 rpm, at 30 °C.

#### 2.4.3. Day 3

The next day, serial dilutions were performed for all the grown cultures, with different concentrations of U, in a microtiter plate.

Further steps were followed in the same way from step 2 to 5 of Day 1 of the plate MIC method (Figure 2A).

#### 2.4.4. Day 4

Growth was observed on the plate, and the MIC value was recorded for each strain. Photographs were taken for the record (Figure 2B).

### 2.5. Spectrophotometric Count Method

This method was performed as the following steps (Figure 3A).

#### 2.5.1. Day 1

Same as step 1 of the broth dilution step.

#### 2.5.2. Day 2

Culture tubes with 5 mL of LB broth were prepared, and different concentrations of U were added in each tube. A test strain grown overnight was inoculated at 0.1 OD in all the prepared tubes.

Further, the growth of the strain can be monitored by a manual spectrophotometric count or by an automatic plate reader. We used an automatic plate reader Bioscreen C system (Growth Curves USA, Piscataway, NJ, USA) to determine the bacterial growth on a timely basis.

We added 300 µL of the previously prepared suspension to the microtiter plate compatible with an automated plate reader. OD was set to take every 3 h of the time interval. Results are shown up to 33 h but may vary according to the growth of the strain.

#### 2.5.3. Day 3

Results were analyzed, and graphs were plotted and compared to control condition (0 mM U).

## 3. Results and Discussion

### 3.1. Screening of Bacterial Strains, Using the Plate MIC Method

The MIC of bacterial isolates against U was assessed, using the plate MIC method, as described in the method section (Figure 1A). After 12 h of the incubation period, we observed that only two bacterial strains, SRS-2-W and SRS-9-S, could grow on the concentration of 6 mM U (Figure 1B). Strain MA-5-S, identified as *Stenotrophomonas* sp., could also grow on the 6 mM U plate upon further incubation for up to 48 h. We noticed that strain MA5 grew only up to 10^−4^ dilution on the control (0 mM U) plate by 12 h of incubation, but by incubating for 48 h, this strain could grow up to 10^−7^ dilution on the control plate (Figure 1B). The results suggest all three bacterial strains are within the same MIC value, but the growth rate of MA-5-S was slower than the other two strains under the test conditions. Similarly, another strain, SRS-11-S, was the slowest to grow, as shown by the growth on the control plate. Upon further incubation to 48 hours, this strain showed growth on a plate with a U concentration of 5 mM (Figure 1B). The MIC was recorded after 48 h of incubation period, as no difference in growth was seen afterward. The following order of the MIC for the tested strains was obtained by using the following method:

SRS-2-W ≥ SRS-9-S ≥ MA-5-S > SRS-11-S > SRS-19-S.

Notably, we obtained similar MIC values upon replicating this assay when performed using the same condition. The only change observed was variability in the microbial growth in their dilution range. For example, growth occurred till 10^−7^ dilution or sometime 10^−6^ or 10^−5^ at one U concentration, which did not result differences in the MIC values obtained. While performing this MIC method, we could also observe the minimum concentration of U when microbes become vulnerable. For example, strain SRS-19-S formed colonies up to 10^−5^ on 0 mM plate, but the addition of 1 mM U impeded its growth by dilution of 10, as the strain grew up to only 10^−4^ dilution; growth was further reduced upon addition of more U (Figure 1B). From these results, we calculated the MIC value and susceptible U concentration, as shown in Appendix A. We also ran the MIC method against several other bacterial species, including *Burkholderia*, *Bacillus*, *Bradyrhizobium*, *Pseudomonas and Paenibacillus sps*. in Appendix A.

We also recorded the colored photographs of plates, as our study focuses on diverse microbes obtained directly from environmental samples. This facilitated the distinction of microbes based on their color and/or morphology. We also observed the interesting trend of the depigmentation of the *Serratia* strain SRS-9-S upon increasing the concentration of U exposure, whereas another *Serratia* sp., SRS-2-W, did not show that characteristic (Figure 1B). Many scientific groups have been focusing on studying the pigmentation property of microbes, including bacteria and fungi [22,23,24,25,26,27,28]. Some of the microbes have been named based on their pigments; e.g., a bacterium named *Chromobacterium violaceum* contains blue–violet pigment [29]. Moreover, pigments have also been found to be associated with microbes’ virulence, and their depigmentation has been demonstrated to decrease their associated virulence; e.g., *Staphylococcus aureus*’s depigmentation by using the gene deletion approach has been correlated with the decrease in its virulence [25,26]. Additionally, the role of microbial pigments has also been shown to be a pharmacological agent; e.g., a pigment, prodigiosin, produced by *Serratia marcescens* has been known to possess cytotoxic activity against many types of cancer cell lines [30,31]. Moreover, the effect of heavy metals on microbial pigmentation has also been probed, suggesting that metal may cause certain alterations in microbial physiology, without affecting their growth [32]. The depigmentation may also be the result of alteration in the electronic properties of the molecule(s) that are the source of the color as a consequence of metal coordinated interaction with these molecule(s). Based on these available studies, we hypothesize that depigmentation of strain SRS-9-S-2018 upon U exposure may correlate with cell viability, physiology, pathogenetic properties, or any U-coordinated interaction. Therefore, a detailed study on pigmentation behaviors will certainly enhance our understanding of undiscovered mechanisms, and our MIC method can be a useful source in such types of primary screening.

### 3.2. MIC Determination, Using Broth Dilution Method, and Comparison to the Plate MIC Method

Further, we compared the plate MIC method to another method of MIC—the broth dilution method. To do so, one of the strains, MA-5-S, was taken, and MIC was performed in a stepwise manner, as described in the method section of broth dilution (Figure 2A). Using this method, we observed that the strain MA-5-S formed colonies at up to 5 mM of U treatment for 24 h, and its growth was completely inhibited at a U concentration of more than 5 mM (Figure 2B), which is a similar result as was seen in the plate MIC method. However, compared to the plate MIC method, this method took us three days to visualize the result and multiple steps were involved. It also required many tubes per strain, thus limiting the number of microbes for screening at the same time.

### 3.3. The MIC Determination, Using Spectrophotometric Count, and Comparison with the Plate MIC Method

In the spectrophotometric count method, a microbial’s growth is calculated based on its optical density, using a spectrophotometer. An increase in the microbial’s growth becomes directionally proportional to an increase in its optical density (OD). To compare this method with the plate MIC method, we selected strain MA-5-S and followed the steps as described in Section 2.5, Spectrophotometric Count Method (Figure 3A). An automated plate reader was used for measuring the optical density in a regular time interval. We noted that cells treated with a U concentration of up to 2 mM were able to grow as appeared by an increase in its OD (Figure 3B,C). At a concentration greater than 2 mM U, the cells did not show any increase in OD (Figure 3B,D).

The OD was found to increase consistently with respect to the increase in U concentration at time 0. The results show the rise in OD, irrespective of the bacterial growth, was caused by the U concentration and poor solubility in the media used. This becomes a concern for the MIC determination when using this method; this method also restricts the number of microbes that can be screened simultaneously, since the plate reader can only accommodate one or two plates. In the case of a manual OD count, the measurements would be time-consuming and inefficient. Thus, the use of the plate MIC method proves to be more advantageous over the spectrophotometric count method.

### 3.4. MIC Determination of a Yeast Strain, Using Plate MIC Method

Furthermore, we applied the plate MIC method on yeast-like fungal strain SRS-4-S-2018 (*Aureobasidium* sp.). The procedure was performed similarly to bacteria, except for the use of the YPD medium instead of the LB medium. The plate MIC of strain SRS-4-S showed that this strain could resist up to 4 mM of U concentration (Figure 4). We could also observe the susceptible concentration of U for this strain in a similar way to bacteria and values shown in Appendix A, along with other tested yeast strains.

### 3.5. MIC Determination of a Fungal Strain, Using Plate MIC Method

We also performed the plate MIC method on filamentous fungal isolates. To do this, a fungal strain was grown on a PDA plate for one week and served as an inoculum for the MIC determination. PDA plates were prepared with different concentrations of U, ranging from 0 to 20 mM. A 5 mm plug of fungus was cut from the fresh growing plate and placed in the center of all the test plates. The fungal growth was monitored by measuring the diameter in a day-wise manner and plotted as a histogram (Figure 5). Using this method, we tested many different fungal sps. We show the data obtained from two fungal strains here, SRS-40-S-2018, and SRS-17-S-2019, which were isolated from a U contaminated site. Strain SRS-40-S-2018 showed higher U resistance as compared to SRS-17-S-2019, as evidenced by their diameter and plotted data (Figure 5).

Additionally, from the decreasing diameter, we could identify a particular U concentration at which the fungal cells started to become susceptible (SRS-40-S-2018 at 16 mM and SRS-64-S-2018 at 10 mM), as evident by the plotted data at day 12 (Figure 5A,B). In contrast to bacteria, we could test only one fungal strain per plate, due to its filamentous growth property. The MIC value and the susceptible concentration have been shown in Appendix A, along with other tested fungal strains.

## 4. Conclusions

We developed a plate MIC method to assess heavy-metal-resistance abilities of multiple microbial isolates, namely bacteria, yeast, and fungi, simultaneously, and in relatively shorter timeframes compared to other known MIC methods. In particular, when uranium is used to screen microbial resistance in liquid growth conditions, the media immediately becomes turbid and eventual OD measurements become biased. Therefore, data that rely on liquid growth measurements to survey microbial U resistance remain potentially flawed, such as those demonstrated in Figure 3. The U turbidity issue served as the rationale to develop a solid plate MIC method, reported herein, which is a significant advancement over the liquid MIC screening technique for heavy metals. We found this plate MIC method to be advantageous over other existing MIC determination methods in terms of labor, time, and space. This new method is a significant step toward the evaluation of metal resistance, a prerequisite to understand metal–microbe interactions, and evaluating metal bioremediation efficiencies of isolated bacterial, yeast, and fungal strains.

## Figures and Tables

**Figure 1 mps-03-00021-f001:**
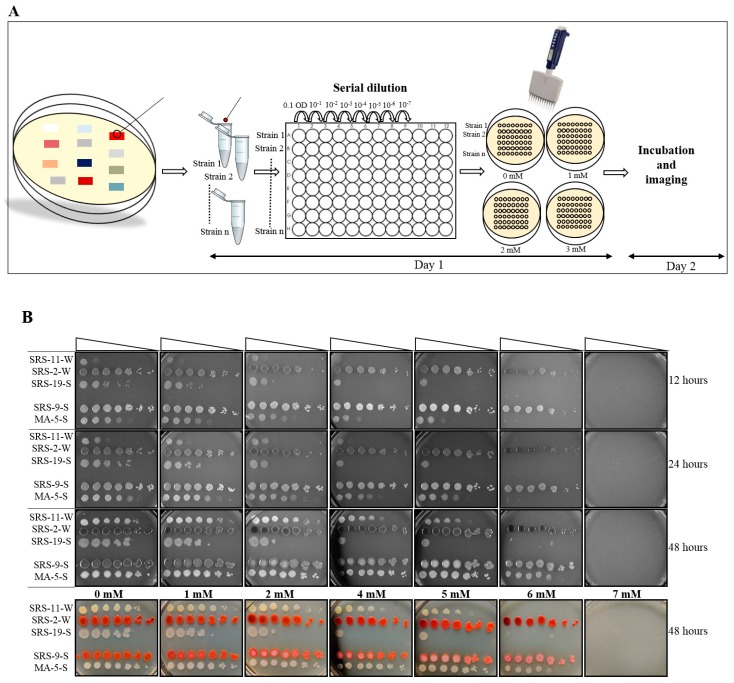
The MIC determination for bacterial isolates against uranium. (**A**) Schematic of plate MIC method. (**B**) Represents the images from plates taken at 12, 24, and 48 h. The U concentration was used from 0 to 7 mM. The procedure was followed as shown in the schematic and described in Materials and Methods, Section 2. The last panel shows colored images at 48 h. Strain names are mentioned on the left of each plate. The experiments was performed in triplicates, and the data shown is a representative image.

**Figure 2 mps-03-00021-f002:**
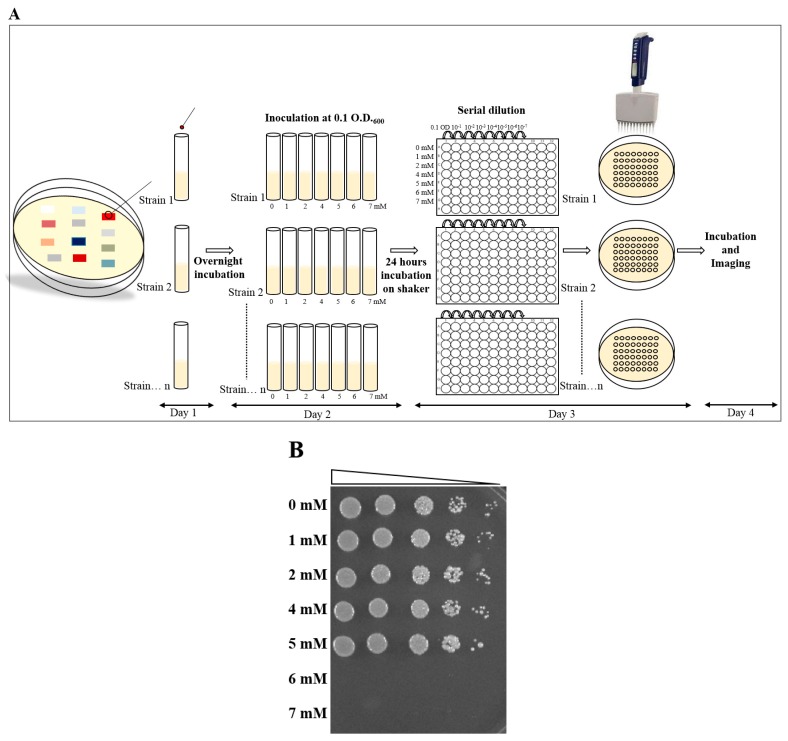
The MIC determination of MA-5-S, using a broth dilution method. (**A**) A schematic shows the steps which were followed in a day-wise manner. (**B**) Represents the plate image taken after 12 h of incubation. The used U concentrations were represented on the left side of the plate.

**Figure 3 mps-03-00021-f003:**
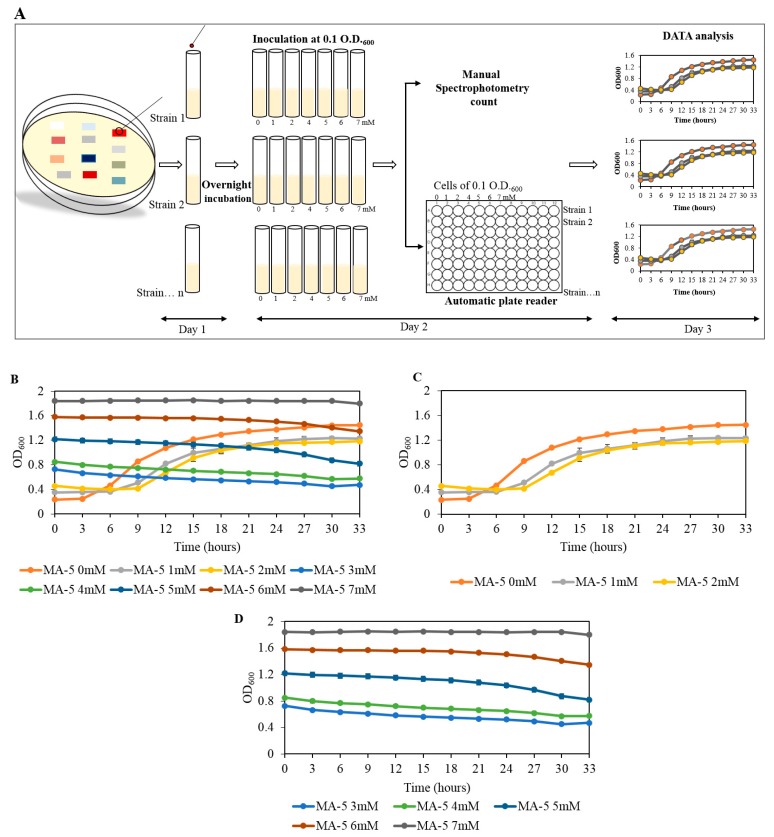
The MIC determination of MA-5, using a spectrophotometric method. (**A**) A schematic shows the steps which were followed in a day-wise manner. (**B**) Represents the growth curve at different concentrations of U. (**C**,**D**) Represents the growth curve at U concentrations ranging from 0 to 2 mM (**C**) and 3 to 7 mM, respectively (**D**).

**Figure 4 mps-03-00021-f004:**
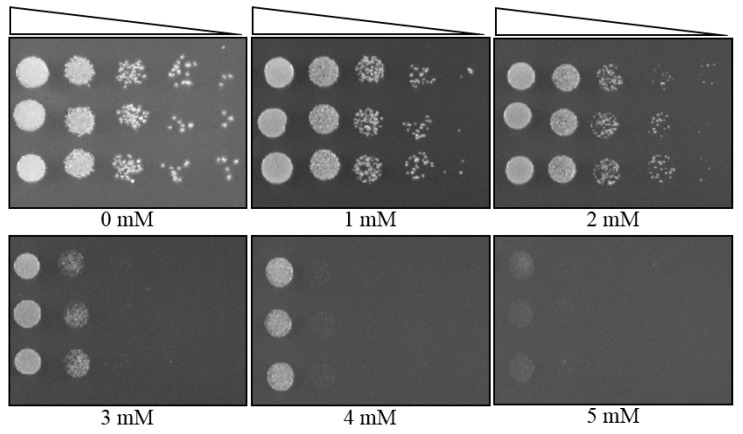
The MIC determination of yeast strain SRS-4-S, using a plate MIC method. All three lanes represent SRS-4-S in triplicate. The steps were followed as described in Figure 1A. Plate images were taken after two days of incubation, at 30 °C.

**Figure 5 mps-03-00021-f005:**
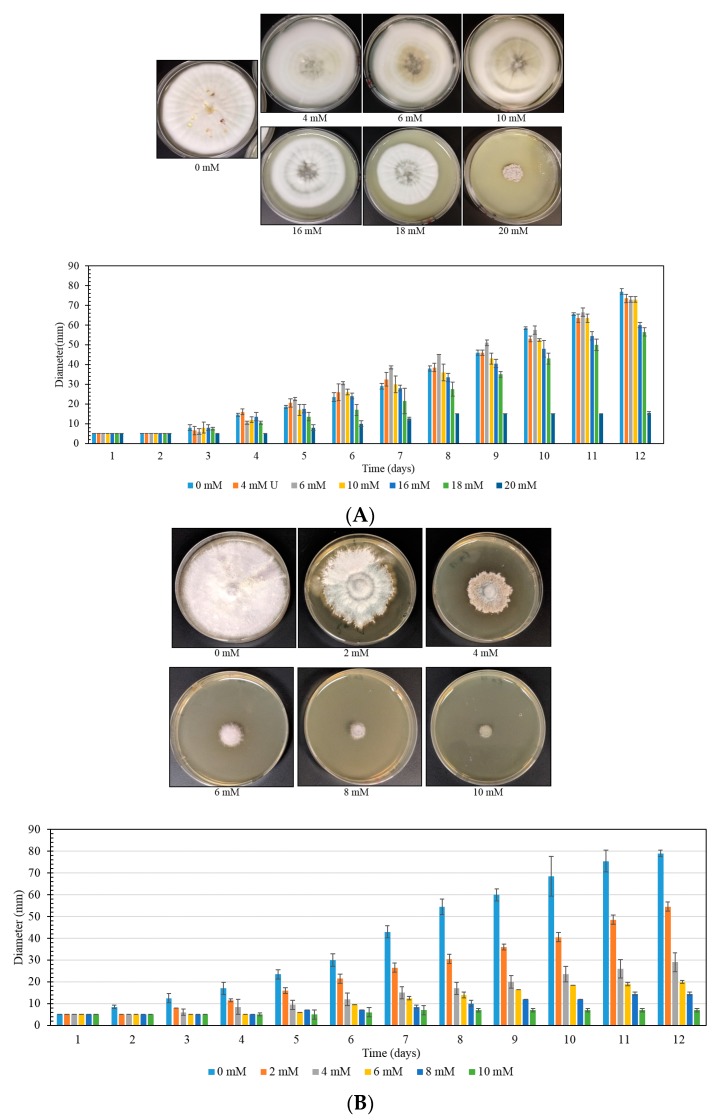
The MIC determination of fungal strains SRS-40-S-2018 (**A**) and SRS-17-S-2019 (**B**), using plate MIC method. Plate images were taken after 12 days of incubation, at 30 °C. The histogram represents the diameter of fungal growth, and measurements were taken on every day, up to 12 days. The experiments were performed in duplicate, and error bars are shown in standard deviation.

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
