# Peer review of "A Rapid and High Throughput MIC Determination Method to Screen Uranium Resistant Microorganisms"

_mps, 2020, doi:10.3390/mps3010021_

Round 1
Reviewer 1 Report
the authors use plate solid method to show that the traditional MIC method, when the media or tested compound affect the OD detection.
It is not a new way to do but maybe for uranium, How and why do you used OD 0.1 as original concentration?
Reviewer 2 Report
I have several major concerns regarding the manuscript that need to be adressed:
For a method establishement the number of strains tested seems insufficient for bacteria and even more so for fungi.
(1) all microorganisms tested should be identified to the species level and not only to the genus level.
(2) number of species tested per group of organisms is not big enough. Dataset should be enlarged as bacteria vary widely in their morphology and growth abilities this should be better refelected in the test set.
(3) Testing one yeast isolate is not sufficient to postulate that the method is applicable for testing a wide range of yeast species
(4) Same applies also for moulds, only 2 representative moulds of the same genus have been tested.
(5) No reference strain is given that was previously tested by other methods and serve as a comparator.
(6) reproducability has not been evaluated to show consistancy of the results, this need to be shown by biological replicates at least 3, assay variability and standard deviation needs to be added, reference strains need to be defined, preferably strains deposit at an international culture collection or type strains
(7) Important information on key measures are missing. Most of all ph of test media which impacts on bioavailability of heavy metals. Also semi- or full-synthetic media are to prefer over complex media such as LB, YPD and PDA. E.g. RPMI is a full synthetic standard medium with a defined ph range.
(8) Differences between media could be eavalauted at least for yeasts and moulds that grow on the same media, why 2 different media were used is not clear.
(9) Figure 5, A end point reading should be choosen before the fungi is covering the whole plate as smaller growth differences might not be observed.
(10) Figure 5, B has major troubles, first of all the authors have seen an Aspergillus niger contamination in the image with 0mM which clearly emerges from the right side, most likely air contamination. Moreover the authors are clearly not dealing with single spore solutions as the fungus grows as multiple sectors like this a diameter evalaution is not possible and accurate.
Reviewer 3 Report
See attached.
